# Environmental Drivers of the Divergence of Harveyi Clade Pathogens with Distinctive Virulence Gene Profiles

**DOI:** 10.3390/microorganisms12112234

**Published:** 2024-11-05

**Authors:** Andrei L. Barkovskii, Cameron Brown

**Affiliations:** Department of Biological and Environmental Science, Georgia College and State University, Milledgeville, GA 30161, USA

**Keywords:** Harveyi clade, divergence, virulence genes, environmental drivers

## Abstract

Fish and shellfish pathogens of the Harveyi clade of the *Vibrio* genus cause significant losses to aquaculture yields and profits, with some of them also causing infections in humans. The present study aimed to evaluate the presence of Harveyi clade fish and shellfish pathogens and their possible diversification in response to environmental drivers in southeastern USA waters. The presence and abundance of potential pathogens were evaluated via the detection and quantitation of six Harveyi-clade-specific virulence genes (*toxR*, *luxR*, *srp*, *vhh_a_, vhh*, and *vhp*; VGs) in environmental DNA with clade-specific primers. The environmental DNA was obtained from water and sediments collected from three Georgia (USA) cultured clam and wild oyster grounds. In sediments, the VG concentrations were, on average, three orders of magnitude higher than those in water. The most and least frequently detected VGs were *vhp* and *toxR*, respectively. In water, the VGs split into two groups based on their seasonal trends. The first group, composed of *luxR*, *vhp*, *vhha*, and *vhh*, peaked in August and remained at lower concentrations throughout the duration of the study. The second group, composed of *toxR* and *srp*, peaked in June and disappeared between July and December. The first group revealed a high adaptation of their carriers to an increase in temperature, tolerance to a wide range of pH, and a positive correlation with salinity up to 25 ppt. The second group of VGs demonstrated a lower adaptation of their carriers to temperature and negative correlations with pH, salinity, potential water density, conductivity, and dissolved solids but a positive correlation with turbidity. No such trends were observed in sediments. These data reveal the role of VGs in the adaptability of the Harveyi clade pathogens to environmental parameters, causing their diversification and possibly their stratification into different ecological niches due to changes in water temperature, acidity, salinity, and turbidity. This diversification and stratification may lead to further speciation and the emergence of new pathogens of this clade. Our data urge further monitoring of the presence and diversification of Harveyi clade pathogens in a global warming scenario.

## 1. Introduction

Many notorious fish and shellfish pathogens belong to the Harveyi clade[1]. Although the composition of the Harveyi clade has been reevaluated over time, most researchers have agreed that it includes *Vibrio harveyi*, *Vibrio parahæmolyticus*, *Vibrio alginolyticus*, *Vibrio campbellii*, *Vibrio diabolicus*, *Vibrio rotiferianus*, *Vibrio owensii*, *Vibrio jasicida*, *Vibrio azureus*, *Vibrio sagamiensis*, *Vibrio natriegens*, and *Vibrio mytili* [2,3]. Their plastic genomes allow *Vibrio* species to rapidly adapt to environmental changes [4]. Due to their preference for warmer temperatures, ocean warming has largely caused their general spread to new regions [5], including pathogenic *Vibrio* species [6]. Sea-surface warming has been related to an increased presence of pathogenic *Vibrio* in the North Atlantic and North Sea, which was linked to an increase in human infections caused by these pathogens [7,8]. In Northern Europe, *Vibrio* spp. wound infections have been positively correlated with heat waves in the last three decades [9]. This increased presence of vibrio pathogens in previously unoccupied regions caused a global rise in vibrio infections, overcoming those of any other waterborne pathogen [10,11].

The major environmental drivers in the marine environment are temperature, pH, and salinity. This triad is also expected to change the most in the global warming scenario. Various pathogenic *Vibrio* species are emerging in response to ocean warming [6]. Elevated temperature stimulated better adaptation of the zoonotic and human pathogen *Vibrio vulnificus* to the host [12]. Elevated temperature caused not only the expansion of vibrio pathogens but also the conversion of commensal and mutualistic *Vibrio* spp. into opportunistic pathogens [13]. Whereas dysregulation of the major adaptive vibrio mechanisms under elevated temperatures and carbon limitation was observed, the expression of genes encoding virulence factors, such as type-3 secretion systems, lytic enzymes, siderophores, motility, and numerous others, was enhanced under these stress conditions [10]. The above results suggested the expansion of vibrio pathogens to new geographical regions, an increase in their pathogenicity, and their possible further speciation.

Members of the Harveyi clade are highly adaptive to elevated water temperature and variations in pH and salinity in their environments [5,6,14,15]. Allen and Finkel [16] reported that *V. harveyi* populations generate genetic diversity and demonstrated that the plasticity of the *V. harveyi* genome allows for rapid adaptation for survival in changing environments. The most studied human pathogen of the clade, *Vibrio parahæmolyticus,* has been observed in various geographical regions, including the southeastern United States [17,18].

Reports on the impact of salinity on Harveyi clade pathogens are contradictory. Low salinity (10–15 ppt) has been linked to increased virulence of Harveyi clade pathogens [19]. A study of the temporal and spatial distributions of *V. parahaemolyticus* along the Hun-Tai River reported moderate positive correlations of its concentration with temperature and salinity [20]. The upregulation of translation mechanisms in *V. harveyi* was observed for many genes under both low- and high-salinity conditions [14].

In general, *Vibrio* species prefer alkaline environments, but the impact of acidification on the survival and virulence of *V. harveyi* was reported to be mitigated by elevated temperature [21]. Some VGs, e.g., *toxR*, enhanced the resistance of *V. parahaemolyticus* to acidic stress [22]. In *V. chloreae*, which is closely related to the Harvey clade, acidic pH promotes hemagglutinin activation [23].

*Vibrio* bacteria in the Harveyi clade exhibit a high level of genomic diversification. Wang et al. [20] reported the population of *V. parahaemolyticus* in the Hun-Tai River to consist of 12 distinctive sequences. Even though the mechanisms underlying speciation in this clade are not yet clearly understood [24], they may involve diversity in the profiles of virulence genes. Particular virulence genes may contribute to the temperature adaptation of the clade species, potentially impacting speciation among these pathogens. Orlíc et al. [25] observed a high diversity of virulence gene profiles in Harvey clade isolates. They also reported that, in summer, the fraction of isolates carrying *vhpA* was three times higher, while the number of isolates carrying any other of the sixteen virulence genes targeted in this study remained the same. RNA-seq analysis demonstrated that temperature and nutrient adaptation are relevant for the pathogenicity of *V. parahaemolyticus* [26]. Thermal stress adaptation may include the upregulation of the expression of virulence genes in Harveyi clade pathogens, mediated by LuxR [27] or other regulatory proteins.

In this study, we attempted to evaluate the temporal and spatial dynamics attributed to the Harveyi clade virulence genes in water and sediments. The environmental DNA obtained from these matrices collected from cultured clams and wild oyster grounds was used for the detection and quantification of Harveyi-clade-specific VGs. In total, six VGs previously reported as Harveyi-clade-specific [28] were targeted in this study. Among these genes were the transmembrane transcriptional regulator *toxR*, the quorum-sensing master regulator *luxR*, the *vhp* metalloprotease, the *vhh* and *vhh_a_* hemolysins, and the serine protease *srp*. Seasonal and spatial variations in VG distribution and concentrations were profiled for temperature, salinity, pH, turbidity, water density, conductivity, and total dissolved solids concentration. This approach did not allow species-specific attribution of VGs but allowed the assessment of their ecological diversification in response to the above drivers at the clade level.

## 2. Materials and Methods

### 2.1. Field Sites and Sampling Events

Three sampling sites were established within the coastal salt marshes offshore of Townsend, Georgia (Figure 1). These sites were selected to provide better coverage of the area and to account for the uneven spatial distribution of Harveyi clade pathogens. Directly north of Fourmile Island, Site 1 was located along the Julienton River at 31°33′34° N, 81°17′16 W. Site 2 was located at the mouth of the Sapelo River at 31°32′33 N, 81°16′53 W. Site 3, the southernmost site, was located along the Mud River at 31°30′21 N, 81°16′44 W. Julienton River, Sapelo River, and Mud River are primarily tidal rivers that feed into the Sapelo Sound Bay. Each site was in direct proximity to cultured clam and wild oyster beds owned and operated by Sapelo Sea Farms. Five sampling events were conducted during low tide in June 2022, August 2022, October 2022, December 2022, and February 2022.

### 2.2. Sample Collection and Processing

Each site was sampled in three locations in triplicate within 20 m of each other to provide variation within the individual sites, providing nine independent samples per site. Three samples of 1 L of water were collected at 1.3–1.5 m at each location within each of the three sites with the LaMotte Water Sampling Set (Chestertown, MD, USA). This approach provided extensive coverage of the study area, allowing generalization on the presence and abundance of the targeted VGs. Samples were transferred into sterile 1 L glass bottles and immediately placed on ice. Once transported to the laboratory, the water samples were filtered using a custom water filtration system with 0.22 µm Millipore nitrocellulose filters (MilliporeSigma, Burlington, MA, USA. Total DNA was extracted using the Qiagen DNEasy^®^ PowerWater kit (Venlo, The Netherlands), and DNA was quantified using a NanoDrop ND-1000 Spectrophotometer (Thermo Fisher Scientific^®^, Waltham, MA, USA). Isolated DNA was stored at −20 °C.

Sediments were collected at the same sites from oyster and clam beds with a LaMotte Bottom Sampling Dredge (Chestertown, MD, USA) into sterile 1.5 mL microcentrifuge tubes within 20 m from each other for each site in triplicate to provide variation within individual sites (9 samples per site). The tubes were immediately transferred to sterile Whirl-Pak^®^ bags and placed over ice. In the laboratory, 250 mg of sediment from each site was used for DNA extraction via the Qiagen DNEasy^®^ PowerSoil Kit (Venlo, The Netherlands). DNA was quantified using a NanoDrop ND-1000 Spectrophotometer (Thermo Fisher Scientific^®^, Waltham, MA, USA), and isolated DNA was stored at −20 °C.

### 2.3. Measurement of Environmental Parameters

A Horiba U-52G Multiparameter Water Quality Meter (Horiba, Irvine, CA, USA) was used to monitor temperature (°C), pH, salinity (ppt), turbidity (NTU), dissolved oxygen (mg/L), total dissolved solids (g/L), conductivity (mS/cm), and potential water density (∂t) at each of the three sites.

### 2.4. Genes Targeted and Generation of Standard Curves

Six VGs previously reported as Harveyi-clade-specific [28] were targeted in this study. These included the transmembrane transcriptional regulator *toxR*, the quorum-sensing master regulator *luxR*, the *vhp* metalloprotease, the *vhh* and *vhh_a_* hemolysins, and the serine protease *srp*. Clade-specific primers previously described by Ruwandeepika et al. [28] were used.

Standard curves were generated with pure cultures of *V. harveyi* (ATCC 14126) and *V. campbellii* (ATCC BAA-1116/BB120). The cultures were grown in Marine Broth overnight at 32 °C, and DNA was extracted with the Qiagen DNEasy Microbial Kit (Venlo, The Netherlands). DNA was quantified using a NanoDrop ND-1000 Spectrophotometer (Thermo Fisher Scientific^®^), subsequently diluted to a concentration of 10 ng/µL, and amplified with a PCR BioRad CFX 1000 (BioRad^®^, Hercules, CA, USA) The assay was optimized to include 10 ng of pure *V. harveyi* or *V. campbellii* DNA, 5 µL of BioRad^®^ PCR Master Mix (Hercules, CA, USA), 3 µL of nuclease-free molecular-grade water, and primers at 250 nM to achieve a total reaction volume of 10 µL. Pure *Escherichia coli* DNA was used as a negative control. The primers used to amplify *toxR*, *luxR*, *srp*, *vhh_a_*, *vhp*, *vhh*, and *rpoA* are shown in Table 1. The cycling parameters were as follows: initial activation, 2 min at 50 °C; denaturation, 10 min at 95 °C; annealing, 20 s at 55 °C; and elongation, 30 s at 72 °C for 45 cycles [28].

The resulting PCR products were transferred to sterile 1.5 mL microcentrifuge tubes and labeled. The products were purified using the Qiagen DNEasy^®^ PowerClean Clean Up Kit (Venlo, The Netherlands) and then quantified using a Qubit Spectrophotometer (Thermo Fisher Scientific^®^). Concentrations were recorded for each target gene product, and the number of gene copies per tube was calculated using the following formula: number of copies = (ng of DNA * (6.022 × 10^23^))/(size of amplicon in bp * (1.0 × 10^9^) * 650). Copy number calculations ranged from 3.90 × 10^10^ to 1.32 × 10^11^, and each DNA sample was serially diluted 10-fold. Detection limits varied by gene, and therefore, the number of serial dilutions varied for each DNA sample.

The diluted DNA for each gene served as a standard and was run in triplicate through a dye-based qPCR assay. The assay was optimized to include 1 µL of DNA at the desired dilution, 5 µL of BioRad^®^ SYBR Green Master Mix (Hercules, CA, USA), 3 µL of nuclease-free molecular-grade water, and 1 µL of forward and reverse primers at 250 nM. The PCR and qPCR assays employed the same primers and cycling conditions. A melt curve step was included at the end of cycling in the RT-qPCR assays. The assays also included a non-template control and serially diluted *E. coli* DNA as a negative control.

Quantitative PCR was performed using BioRad CFX 1000 (BioRad^®^). The MASTERO BioRad software (Hercules, CA, USA) employs an automated linear regression feature to describe the relationship between the log of the copy number values and the cycle threshold (Ct) values for each assay. The equation provided for each gene’s standard curve was further used to calculate the number of gene copies in environmental samples.

### 2.5. Statistics

Virulence genes were unevenly distributed between the sites. To better reflect their presence and concentrations within the study area, concentrations of each VG observed in sites 1–3 were separately averaged for water and sediment in every sampling event. Variations in gene concentrations among sites were evaluated with Welch’s ANOVA, which revealed that there were no statistically significant differences in average gene copies among sites 1–3. The values for error bars were too low to be presented in logarithmic-scale figures, so these are not included in the figures. Pearson’s correlation coefficient was calculated and applied to evaluate the relationship between gene concentrations and water parameters. Obtained correlations were tested on their statistical significance using the *P*-test.

## 3. Results

### 3.1. The Presence and Seasonal Distribution of Virulence Genes

Each VG was detected at least once during this study, but the gene profiles varied between water and sediments and from site to site. Their copy numbers varied considerably among sampling events and between water and sediments. All six VGs were detected in water samples in the summertime (Table 2).

In water, the highest copy numbers were observed in June and August for *toxR* at roughly 8.0 × 10^5^ and 8.0 × 10^3^ and for *vhha* at 3.0 × 10^3^ and 4.9 × 10^4^, respectively. In sediments, the highest copy numbers were detected for *vhh_a_* and *toxR* in the same months roughly at 5.0 × 10^8^ and 1.0 × 10^7^, respectively. Despite fluctuations in gene concentrations, no site-related trends were observed. To better reflect their presence and concentrations within the study area, each VG’s observed concentrations in sites 1–3 were separately averaged for water and sediment in every sampling event. Variations in average gene concentrations among the sites were evaluated with Welch’s ANOVA, which revealed no statistically significant differences. The values for error bars were too low to be presented in logarithmic-scale figures, so these are not included in the figures (Figure 2 and Figure 3).

Of the six VGs detected in water, only two, *luxR* and *vhp*, persisted in all five sampling events. Three genes, *vhh_a_*, *srp*, and *vhh*, were detected at four sampling events. The other, *toxR*, was only detected in June and August. Gene concentrations widely varied in the range of 1.00 × 10^−2^ to 8.0 × 10^5^ copies/mL between individual genes and seasons (Figure 2). In water, the VGs split into two groups on their seasonal trends, with the first group (endurable, E-group), composed of *luxR*, *vhp*, *vhha*, and *vhh*, peaking in August and remaining at lower concentrations for the duration of the study. The second group (perishable, P-group), composed of *toxR* and *srp*, peaked in June and disappeared between July and December (Figure 2).

In sediments, VG concentrations were generally higher than in water (Figure 3). They varied between individual genes in the range of 1.00 × 10^1^ to 1.00 × 10^9^ copies/g (Figure 3). All the VGs were observed in all five sampling events, and their concentrations varied less than those in water. All the VGs revealed a similar seasonal trend, with higher concentrations in June and August, followed by a decline through December. A high abundance and persistence of *toxR* and *srp* in sediments through the study suggest that sediments are the primary habitat for their carriers in fall and winter. The temporal distribution of VGs did not reveal their divergence into two groups in sediments.

### 3.2. Divergence of VG Carriers in Response to Water Parameters

Pearson’s coefficient was used to analyze correlations between water parameters and VG concentrations. VG concentrations were related to pH, temperature, salinity, turbidity, dissolved oxygen, conductivity, total dissolved solids, and potential water density, whose values are presented in Table 3.

In water, the concentrations of the E-group, composed of *vhh*, *vhh_a_*, *vhp*, and *luxR* genes, were strongly positively correlated with temperature, moderately with salinity, and yet weaker with conductivity and total dissolved solids. Their correlations with pH and dissolved oxygen were moderately negative, except for *vhp*, and slightly negative with turbidity and potential water density (Table 4).

Their concentration correlations with the temperature and pH were statistically significant. Correlations with the other parameters were either of lower statistical significance or insignificant.

Except for the dissolved oxygen concentration, to which both groups reacted similarly, the VGs in the P-group of genes (composed of *toxR* and *srp*) demonstrated an almost inverse trend. In contrast to the E-group, the concentrations of the P-group of genes were only moderately positively correlated with temperature but highly negatively with pH. In contrast to the E-group, concentrations of the P-group of genes were negatively correlated with salinity, conductivity, and total dissolved solids, and also with potential water density. Salinity, conductivity, total dissolved solids, and potential water density largely reflect the sum concentration of polar inorganic and organic compounds that, in marine environments, are primarily salts and organic acids. In the meantime, their concentrations were positively correlated with turbidity, with which the genes of the E-group correlated negatively (Table 5). All of the above correlations were statistically significant.

The inverse relations of the E- and P-groups of VGs with the major water parameters suggest different carriers for these two groups of genes. The positive correlation of the P-group of VGs with turbidity may indicate an association of their carriers with suspended particles, which is in agreement with their significantly higher concentrations and persistence in sediments compared to water.

To better reflect the environmental fate of E- and P-gene carriers, the average concentrations of VGs in each group in response to environmental drivers are presented in Figure 4, Figure 5 and Figure 6. The dots in these figures represent averaged values for E- and P-groups of genes and water parameters for each sampling event. Since both values were averaged for sites 1–3 and the logarithmic Y scale was used to accommodate the concentration range observed in the experiment, no error bars are included. The actual water parameters of each site recorded at each sampling event can be referred to in Table 3.

The average concentrations of VGs in the E-group were positively correlated with the temperature in the range of 10–30 °C (Figure 4A), whereas the concentrations of VGs in the P-group decreased after the temperature reached 26 °C (Figure 4B). The E-group revealed two concentration peaks at around pH 7.15 and pH 7.45 (Figure 5A), whereas the highest concentration of the P-group of genes was observed at pH 7.10, with a steady decline thereafter and complete disappearance above pH 7.40 (Figure 5B).

Both groups also diverged in their reactions to salinity. The average concentration of the E-group steadily increased between 19 and 24.5 ppt of salts (Figure 6A), and the concentrations of the P-group of genes decreased in the same salinity range (Figure 6B). The concentration dynamics of both groups in response to other water parameters also exhibited inverse trends. Overall, the two groups exhibited opposite trends in reaction to most of the water parameters. This observation suggests that the virulence genes are possible regulators for the divergence of the Harveyi clade pathogens and their ecological diversification in the marine environment. It also suggests the diversification of Harvey clade pathogens based upon the profiles of virulence genes in response to water parameters as environmental drivers.

This divergence pattern was not observed in sediments (Table 4). All the genes had a strong positive correlation with the temperature. Three genes, *toxR*, *luxR*, and *vhp,* had weak positive correlations with the salinity, conductivity, and total dissolved solids, while *srp*, *vhh*, and *vhh_a_* had weak negative correlations. Additionally, *toxR*, *luxR*, and *vhp* demonstrated a weaker negative correlation with potential water density than the other genes. In contrast to the distinct patterns observed in water samples, there was no obvious separation between the two VG groups in their reaction to water parameters in sediments.

## 4. Discussion

Vibrio bacteria in the Harveyi clade exhibit a high level of genomic divergence, but the mechanisms underlying speciation in this clade have not yet been identified [24]. In this study, we attempted to evaluate whether different VG profiles cause ecological diversification among the *Vibrio* Harveyi clade pathogens in response to environmental drivers. Rather than evaluate the spatial and temporal diversity of clade species, we proposed signature virulence genes [28] to address the above diversification. In this case, we assumed that these genes are not equally distributed between all the pathogens of the clade; that various combinations of these genes impact the fitness of their carriers; and that, therefore, their presence and concentrations may reflect the adaptivity of their carriers to cardinal water parameters, predicting the possible divergence of these pathogens in a global warming scenario.

In this study, we demonstrated that the divergence of Harveyi clade pathogens may involve variances in the VG profiles of their carriers and the response of the latter to environmental factors.

The six virulence genes targeted in this study (*toxR*, *luxR*, *srp*, *vhh_a_*, *vhp*, and *vhh*) have been well documented in pathogenic species belonging to the *Vibrio* Harveyi clade, and the primers used in this study were selective for the Harveyi clade pathogens [28]. While their presence does not necessarily indicate the high acute virulence of their carriers, it does indicate the presence of Harveyi clade pathogens, and their concentrations reflect the abundance of these pathogens.

VG concentrations were significantly higher in sediments than in water, indicating greater persistence of Harveyi clade *Vibrio* in sediments. Sediments protect against environmental stressors, so these results were unsurprising [29,30]. These findings are consistent with a recent study where the concentrations of VGs and tetracycline resistance genes in sediments were up to three orders of magnitude higher than those in water [31]. Böer et al. [32] reported *Vibrio* spp. concentration in sediments up to three orders of magnitude higher than in water [32], which is also consistent with the results of this current study.

The generally observed seasonal trend of the VG concentrations being higher in summer in both water and sediments was expected [19,33]. Since *Vibrio* spp. have a preference for warmer waters and exhibit optimal growth at 20–35 °C, gene copy numbers were expected to be greater during summer than winter [34]. Previous studies on *Vibrio parahaemolyticus* in the region of the current study have also shown similar seasonal trends [18].

Of particular interest, however, is the evidenced heterogeneity in the distribution of these genes in response to water parameters. This heterogeneity was more pronounced in water than in sediments, likely due to the mitigating role of sediments against changes in water parameters [29,30]. The distribution of VGs among Harvey clade carriers has been shown to be largely scattered. A recent study reported that only 10.8% of all Harvey clade isolates carried more than one of sixteen targeted VGs [25], suggesting that the random distribution of VGs among Harveyi clade carriers is not necessarily species-specific and may indicate the divergence of their carriers in response to multiparametric changes in their environment. Global warming is altering the pH, temperature, and salinity of the world’s oceans at a global scale, which impacts the pathogenicity of *Vibrio* species [35] and warrants a study on the relationships between VG profiles of the Harveyi clade pathogens and water parameters, and possibly on their role in the diversification of their carriers in response to major environmental drivers.

We observed the diversification of VG patterns among Harveyi clade pathogens in response to temperature. The abundance of the E-group of genes was highly correlated with elevated temperatures, whereas that of the P-group was not. Previously, elevated temperature has been shown to correlate with increased virulence gene expression and pathogenicity of Vibrio carriers [12,26,27]. A positive correlation between the VG copy numbers associated with *V. parahaemolyticus* and *V. vulnificus* and temperature was also reported [18,33]. Particular VGs provide adaptive advantages to Harvey clade pathogens under elevated temperature conditions. Orlíc et al. [25] reported a disproportional increase in *vphA* carriers among other Harvey clade pathogens in the summer. Higher temperatures also caused a significantly higher number of *V. parahaemolyticus* isolates carrying *tdh* and *trh* VGs [33]. This adaptive advantage to elevated temperature could be a result of multiple high-frequency mutations of particular VGs, as was shown for the *tlh* gene in *V. parahaemolyticus* [15]. Thermal stress adaptation may also include the upregulation of VG expression in the Harveyi clade pathogens, mediated by LuxR [27] or other regulatory proteins. In contrast, elevated temperature may also cause the dysregulation of the major adaptive vibrio mechanisms, leading to a complex response of Harvey clade pathogens to temperature [10]. The heterogeneity of VGs in E- and P-groups observed in the current study suggests the adaptive advantage of carrying particular VGs in response to temperature. This also suggests the divergence of the Harveyi clade carriers based upon VG profiles under the selective pressure of environmental drivers.

The divergence of the VGs and their Harvey clade carriers into two groups in our study was driven not only by temperature but also by pH. In general, *Vibrio* species prefer alkaline environments, but the intra- and interspecies variability of *Vibrio* spp. in relation to pH has been previously reported, with optimal values ranging between slightly acidic and highly alkalic [36]. The E-group of genes reached the highest concentration peaks at around pH 7.15 and 7.45, whereas the highest concentration of the P-group of genes was observed at pH 7.10, followed by a sharp decline. A recent study on the impact of pH on the growth of *V. parahaemolyticus* and *V. vulnificus* demonstrated that this impact was non-linear, with maximum growth rates at approximately pH 5.5, pH 7.0, and pH 8.2 in planktonic cultures and more randomized growth patterns in biofilms [36]. The authors also reported that an interplay between pH and temperature caused shifts in growth patterns. Another study demonstrated that the impact of acidification on the survival and virulence of *V. harveyi* was mitigated by elevated temperature [21].

Lower pH has also been reported to cause higher virulence of *Vibrio* species. In *V. chloreae*, acidic pH promoted hemagglutinin activation [23]. The impact of pH on the production of virulence factors was observed in *V. cholerae,* and the evolution of its ability to respond to novel signals was suggested [37]. A particular VG, *toxR*, was previously reported to be required for the survival of *V. parahaemolyticus* under acidic stress conditions [22]. In our study, *toxR* was one of the two genes in the P-group with the highest concentration at the most acidic conditions, confirming the positive selection for the Harveyi clade carriers of this gene under low-pH conditions.

Salinity was another factor that contributed to the divergence of Harveyi clade pathogens. The carriers of E- and P-groups of genes revealed the opposite trends in relation to salinity. Harveyi clade species are halophilic and require around 15–25 ppt of salts in their environment [5,6]. The optimal salinity for *V. parahaemolyticus* was reported to be between 15 and 30 ppt [38]. The optimal salinity for *V. harveyi* was reported to be between 15 and 35 ppt [39]. A study on the impact of salinity on *V. harveyi, V. parahaemolyticus,* and *V. vulnificus* suggested that the presence of particular VGs was correlated with the growth of the carriers under high-salinity conditions [20]. On the other hand, Kloska et al. [14] demonstrated that the adaptation of *V. harveyi* to both low- and high-salinity conditions required very similar upregulation changes in the transcription of housekeeping genes. The relationships of *Vibrio* species with salinity are complex and change with the salinity range. This was clearly illustrated in a study on the impact of salinity on *V. vulnificus* [40]. An inverse correlation of *V. vulnificus* abundance with salinity with an optimum at 5 to 10 ppt was the general trend in this study. Nevertheless, in the range of 20 to 25 ppt, *V. vulnificus* abundance demonstrated a positive correlation with salinity. This study suggests that the divergence of Harvey clade pathogens in their relationships to salinity may occur not only at the interspecies level but also at the intraspecies level.

The E- and P-groups of genes demonstrated the opposite trends for turbidity. Previous studies observed a positive correlation between turbidity and the density of *V. parahaemolyticus* populations in water and sediments, suggesting sediment disturbance and the simultaneous release of particle-associated pathogens as the driving force [11,41]. The carriers of the P-group were mostly associated with sediments in our study, confirming the previous suggestions and explaining their positive correlation with turbidity.

In contrast, Scro et al. [42] reported that high turbidity was associated with a decrease in *Vibrio* spp. concentration in water and sediments. This observation may explain the negative correlation of the E-group of genes with turbidity in the current study. Overall, our and others’ studies indicate that the divergence of *Vibrio* spp. might be driven by turbidity.

In conclusion, our study reveals the role of VGs in the adaptability of the Harveyi clade pathogens to environmental parameters, causing their diversification and possibly their stratification into different ecological niches, mainly due to changes in water temperature, acidity, salinity, and turbidity. Alone or in combination, these environmental drivers may cause temporal and spatial stratification of Harveyi clade pathogens across temperature, pH, salinity, and turbidity gradients, possibly accelerating the speciation of the Harveyi clade. Such speciation may cause the emergence of new pathogenic Harveyi clade strains and species in a global warming scenario and increased anthropogenic impact on coastal marine waters.

## Figures and Tables

**Figure 1 microorganisms-12-02234-f001:**
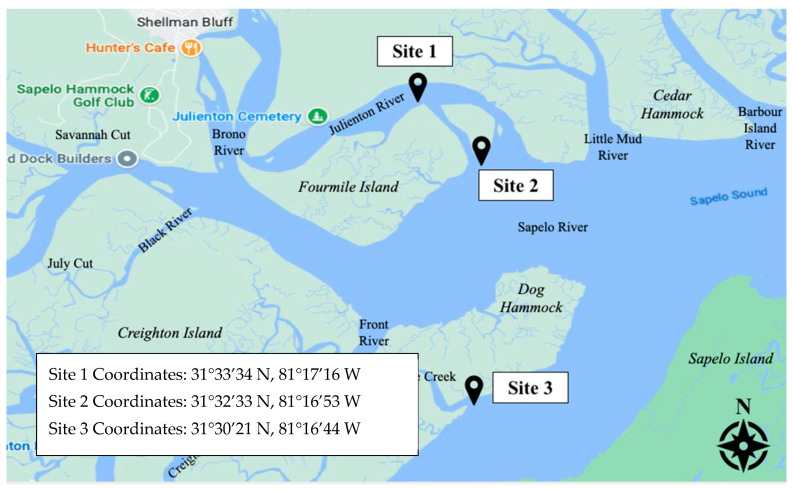
A map of the three study sites. Each site was in direct proximity to cultured clam and wild oyster beds. The sites were selected for better coverage of the study area.

**Figure 2 microorganisms-12-02234-f002:**
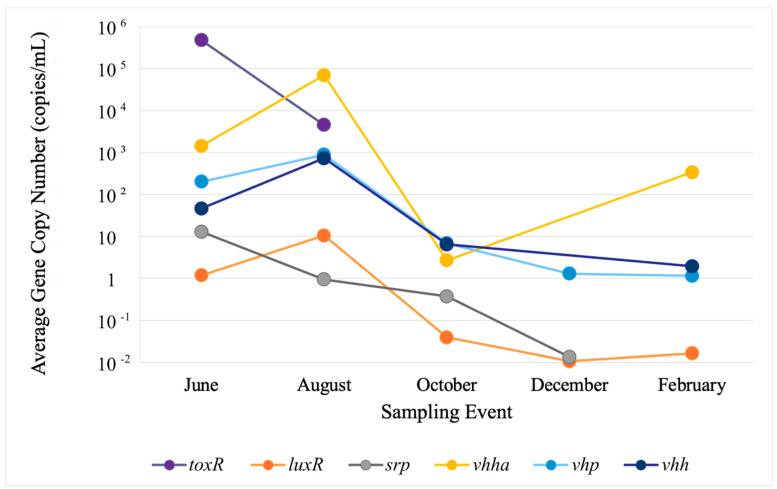
Average VG copy numbers in water. To better reflect their presence and concentrations within the study area, concentrations of each VG observed in sites 1–3 were averaged for every sampling event. Variations in average gene concentrations among sites were evaluated with Welch’s ANOVA, which revealed no statistically significant differences. The values for error bars were too low to be presented in logarithmic-scale figures, so these are not included.

**Figure 3 microorganisms-12-02234-f003:**
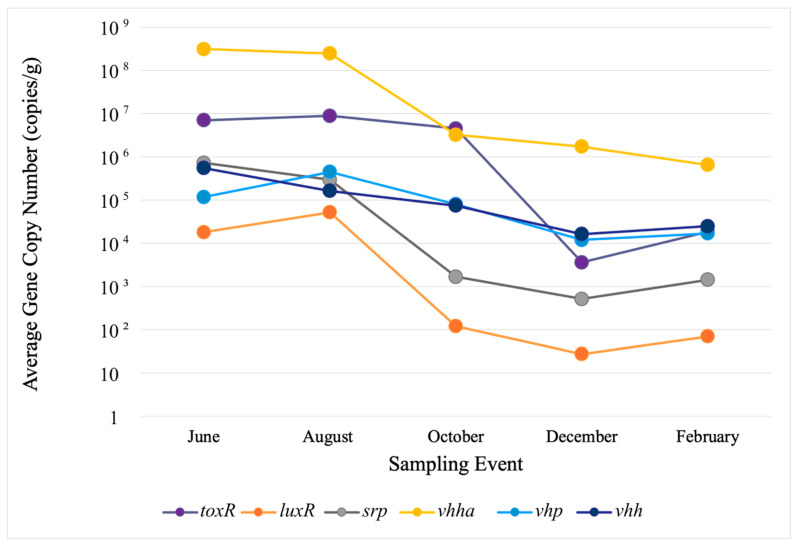
Average VG copy numbers in sediments. To better reflect their presence and concentrations within the study area, concentrations of each VG observed in sites 1–3 were averaged for every sampling event. Variations in average gene concentrations among sites were evaluated with Welch’s ANOVA, which revealed no statistically significant differences. The values for error bars were too low to be presented in logarithmic-scale figures, so these are not included.

**Figure 4 microorganisms-12-02234-f004:**
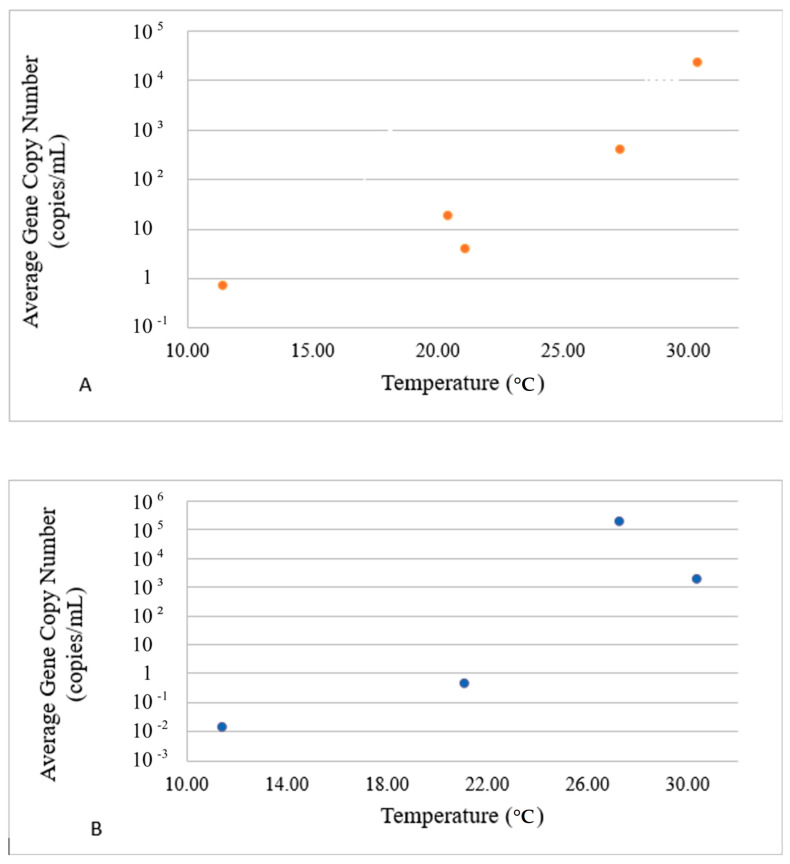
Impact of temperature on the presence of E ((**A**), top panel) and P ((**B**), bottom panel) VG groups. The dots in these figures represent averaged values for E- and P-groups of genes and water parameters for each sampling event. Since both values were averaged for sites 1–3 and the logarithmic Y scale was used to accommodate the concentration range, no error bars are included.

**Figure 5 microorganisms-12-02234-f005:**
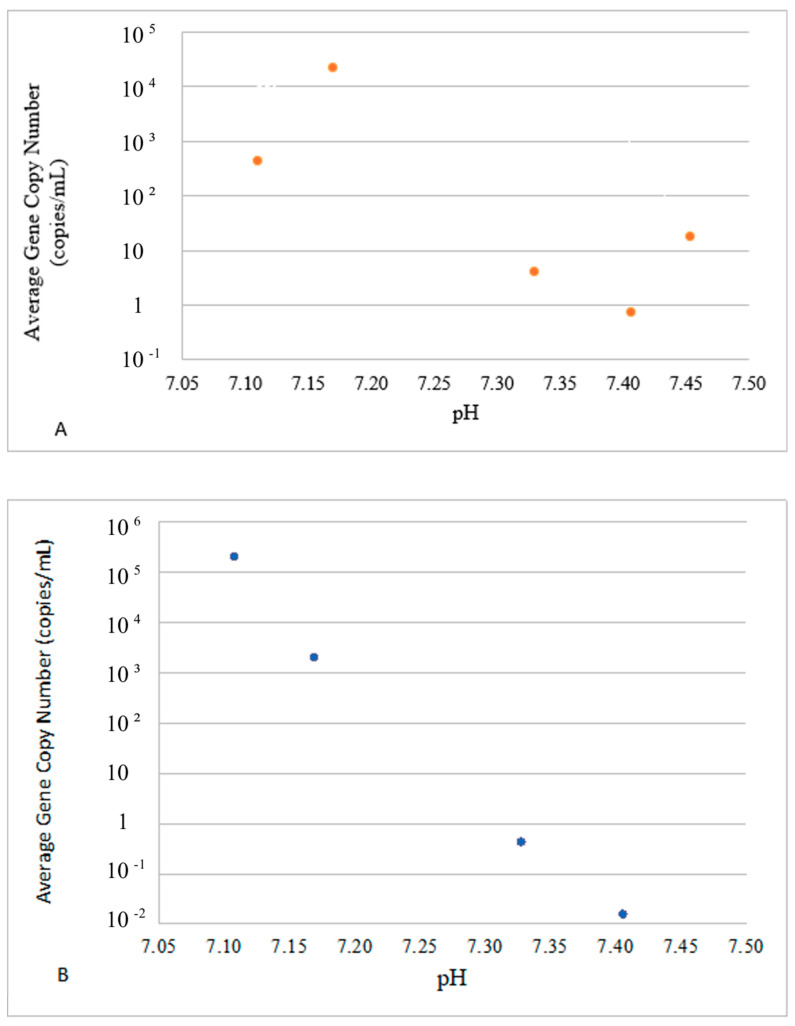
Impact of pH on the presence of E ((**A**), top panel) and P ((**B**), bottom panel) VG groups. The dots in these figures represent averaged values for E- and P-groups of genes and water parameters for each sampling event. Since both values were averaged for sites 1–3 and the logarithmic Y scale was used to accommodate the concentration range, no error bars are included.

**Figure 6 microorganisms-12-02234-f006:**
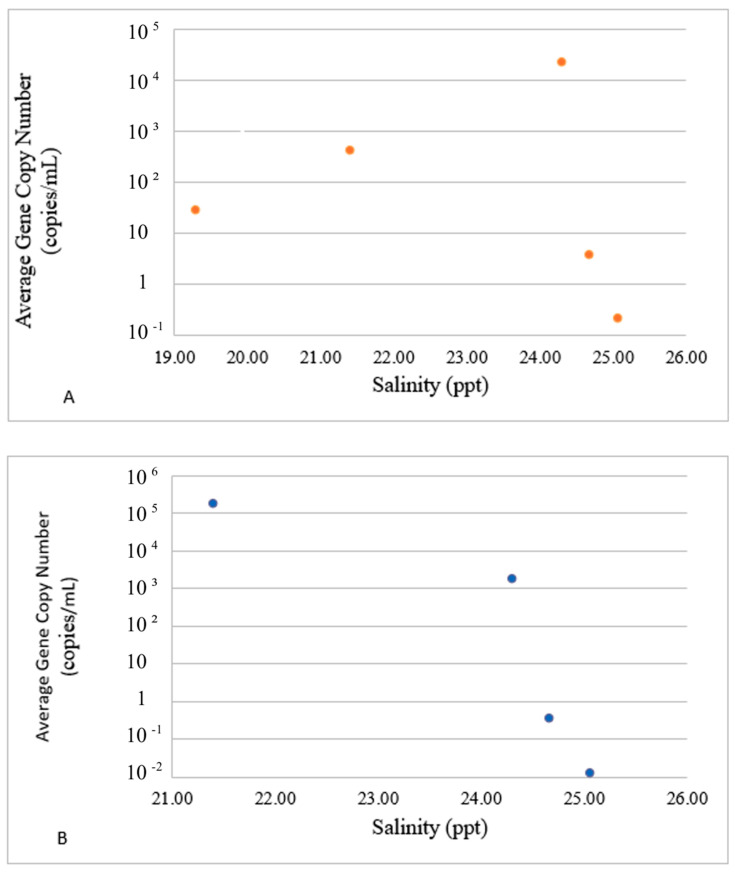
Impact of salinity on the presence of E ((**A**), top panel) and P ((**B**), bottom panel) groups. The dots in these figures represent averaged values for E- and P-groups of genes and water parameters for each sampling event. Since both values were averaged for sites 1–3 and the logarithmic Y scale was used to accommodate the concentration range, no error bars are included.

**Table 1 microorganisms-12-02234-t001:** Target gene primer sequence and amplicon size.

Gene	Primer Sequence	Amplicon Size (bp)	Reference
*luxR*	FW: TCAATTGCAAAGAGACCTCG RV: AGCAAACACTTCAAGAGCGA	84	[28]
*toxR*	FW: CGACAACCCAAAATACGGAA RV: AGAGCAATTTGCTGAAGCTA	131	[28]
*srp*	FW: TGCACGACCAGTTGCTTTAG RV: AAGTGGTCGTCAGCAAATCC	232	[28]
*vhh*	FW: TTCACGCTTGATGGCTACTG RV: GTCACCCAATGCTACGACCT	234	[28]
*vhh_a_*	FW: GCGCTTGGTATCTTCTCTGA RV: CAGACAGCTCATCACGCATT	226	[28]
*vhp*	FW: CTGAACGACGCCCATTATTT RV: CGCTGACACATCAAGGCTAA	201	[28]

**Table 2 microorganisms-12-02234-t002:** Virulence gene occurrence and instance of detection at sites 1–3. The data are based on three independent subsamples (each in triplicate, 9 samples per site) from each site.

Site/Gene	*toxR*	*luxR*	*srp*	*vhh_a_*	*vhp*	*vhh*
Site 1						
June Sediment	+	+	+	+	+	+
August Sediment	-	+	+	+	+	+
October Sediment	+	-	+	+	+	-
December Sediment	-	+	+	+	+	+
February Sediment	+	+	+	+	+	+
June Water	+	+	+	+	+	+
August Water	-	+	-	+	+	+
October Water	-	+	+	+	+	+
December Water	-	+	-	-	+	-
February Water	-	+	-	-	+	+
Site 2						
June Sediment	+	-	+	+	+	-
August Sediment	+	+	+	+	+	+
October Sediment	+	+	+	+	+	+
December Sediment	-	-	+	+	-	+
February Sediment	-	+	+	+	+	+
June Water	+	+	+	+	+	+
August Water	+	-	+	+	+	+
October Water	-	+	+	+	-	+
December Water	-	+	+	-	-	-
February Water	-	+	-	-	+	-
Site 3						
June Sediment	-	+	+	+	-	+
August Sediment	+	+	+	+	+	+
October Sediment	+	+	+	+	+	+
December Sediment	+	+	+	+	+	+
February Sediment	+	+	+	+	+	+
June Water	+	+	+	+	+	+
August Water	+	+	+	+	+	+
October Water	-	+	+	-	+	+
December Water	-	-	+	-	+	-
February Water	-	-	-	+	+	+
Detection Frequency	15/30	24/30	25/30	24/30	26/30	24/30
Sediment Total	10/15	12/15	15/15	15/15	13/15	13/15
Water Total	5/15	12/15	10/15	9/15	13/15	11/15

**Table 3 microorganisms-12-02234-t003:** Water parameters and their averages at three study sites between June 2022 and February 2023.

Water Parameters	pH	Temp. (°C)	Salinity (ppt)	Turbidity (NTU)	DO (mg/L)	Conduct. (mS/cm)	TDS (g/L)	Potential Water Density
June								
Site 1	7.10	27.63	18.3	23.6	1.24	31.5	18.1	10.2
Site 2	7.23	26.33	22.4	21.1	1.18	35.7	22.9	11.4
Site 3	7.00	28.05	23.5	22.5	1.26	38.3	23.4	13.6
Average	7.11	27.34	21.40	22.40	1.23	35.17	21.47	11.73
August								
Site 1	7.11	29.42	23.8	15.7	1.50	38.8	24.4	13.8
Site 2	7.10	31.05	24.2	18.4	1.43	39.4	25.1	13.9
Site 3	7.30	30.80	24.9	17.3	0.87	40.0	24.4	14.1
Average	7.17	30.42	24.30	17.13	1.27	39.40	24.63	13.93
October								
Site 1	7.28	20.65	24.3	18.9	2.54	39.5	25.3	15.8
Site 2	7.33	20.95	24.6	20.2	2.30	39.6	25.6	17.3
Site 3	7.37	21.85	25.1	20.3	2.90	39.0	26.1	16.2
Average	7.33	21.15	24.67	19.8	2.58	39.37	25.67	16.43
December								
Site 1	7.43	11.53	23.8	20.4	3.6	43.8	26.7	20.2
Site 2	7.43	11.52	26.6	25.2	3.51	43.5	26.9	20.0
Site 3	7.36	11.32	24.8	21.7	3.22	40.4	24.4	18.1
Average	7.41	11.46	25.07	22.43	3.44	42.57	26.00	19.43
February								
Site 1	7.52	21.29	19.5	17.6	1.5	33.0	20.1	12.8
Site 2	7.37	20.38	21.4	11.2	0.9	35.6	21.7	14.5
Site 3	7.47	19.54	17.0	7.0	1.98	29.6	18.4	11.3
Average	7.45	20.40	19.30	11.93	1.46	32.73	20.07	12.87

**Table 4 microorganisms-12-02234-t004:** A heat map depicting the correlations between VG copy numbers and water parameters in water samples. *p*-values are presented in brackets.

Gene/ Parameter	pH	Temp. (°C)	Salinity (ppt)	Turbid. (NTU)	DO (g/L)	Cond. mS/cm	TDS (g/L)	Potent. Water Density
*toxR*	−0.697 (0.039)	0.403 (0.136)	−0.344 (0.209)	0.465 (0.081)	−0.443 (0.098)	−0.385 (0.156)	−0.440 (0.101)	−0.574 (0.025)
*luxR*	−0.555 (0.042)	0.692 (0.043)	0.271 (0.329)	−0.156 (0.579)	−0.474 (0.074)	0.185 (0.509)	0.180 (0.521)	−0.240 (0.389)
*srp*	−0.738 (0.002)	0.450 (0.092)	−0.320 (0.245)	0.467 (0.079)	−0.469 (0.078)	−0.371 (0.173)	−0.420 (0.119)	−0.588 (0.021)
*vhh_a_*	−0.480 (0.070)	0.644 (0.009)	0.294 (0.288)	−0.200 (0.475)	−0.428 (0.111)	0.213 (0.446)	0.214 (0.444)	−0.186 (0.507)
*vhp*	−0.643 (0.097)	0.747 (0.001)	0.235 (0.399)	−0.101 (0.720)	−0.530 (0.042)	0.142 (0.613)	0.132 (0.639)	−0.310 (0.261)
*vhh*	−0.515 (0.049)	0.668 (0.007)	0.287 (0.299)	−0.179 (0.523)	−0.449 (0.093)	0.203 (0.468)	0.202 (0.470)	−0.210 (0.453)

Correlations from the highest positive to the highest negative.

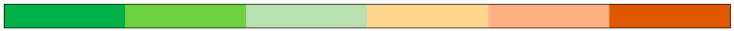

**Table 5 microorganisms-12-02234-t005:** A heat map depicting the correlations between VG copy numbers and water parameters in sediment samples. *p*-values are presented in brackets.

Gene/ Parameter	pH	Temp. (°C)	Salinity (ppt)	Turbid. (NTU)	DO (g/L)	Cond. mS/cm)	TDS (g/L)	Potent. Water Density
*toxR*	−0.927 (0.0001)	0.893 (0.0001)	0.221 (0.429)	0.227 (0.416)	−0.584 (0.226)	0.029 (0.918)	0.098 (0.728)	−0.472 (0.076)
*luxR*	−0.724 (0.023)	0.791 (0.004)	0.188 (0.502)	−0.045 (0.873)	−0.583 (0.226)	0.092 (0.744)	0.074 (0.793)	−0.380 (0.162)
*srp*	−0.897 (0.0001)	0.666 (0.007)	−0.228 (0.414)	0.390 (0.151)	−0.619 (0.014)	−0.302 (0.274)	−0.359 (0.189)	−0.654 (0.008)
*vhh_a_*	−0.957 (0.0001)	0.809 (0.0003)	−0.092 (0.744)	0.278 (0.316)	−0.689 (0.004)	−0.185 (0.509)	−0.235 (0.399)	−0.635 (0.010)
*vhp*	−0.653 (0.083)	0.772 (0.0007)	0.292 (0.291)	−0.087 (0.758)	−0.510 (0.052)	0.169 (0.547)	0.193 (0.491)	−0.291 (0.293)
*vhh*	−0.852 (0.0006)	0.600 (0.018)	−0.251 (0.367)	0.440 (0.100)	−0.559 (0.030)	−0.335 (0.194)	−0.370 (0.175)	−0.636 (0.010)

Correlations from the highest positive to the highest negative.

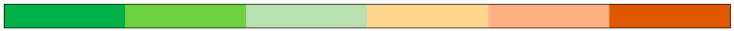

## Data Availability

The original contributions presented in the study are included in the article, further inquiries can be directed to the corresponding author.

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
