# Peer review of "Environmental Drivers of the Divergence of Harveyi Clade Pathogens with Distinctive Virulence Gene Profiles"

_microorganisms, 2024, doi:10.3390/microorganisms12112234_

Round 1
Reviewer 1 Report
Comments and Suggestions for Authors
Overall, the manuscript is limited, although the subject is of good relevance to science. Many sentences should be carefully revised for a better understanding by the reader. Please, see the attached document comments to improve the text.

Comments on the Quality of English Language
The English language is not bad. However, it is important to adequate and better link the sentences.
Reviewer 2 Report
Comments and Suggestions for Authors
The manuscript entitled “Temperature, pH, and Salinity as Environmental Drivers for Divergence Among Harveyi Clade Pathogens and First Evidence for Vibrio harveyi and Vibrio campbellii in Southeastern USA Waters” evaluated divergence among the Vibrio Harveyi clade
pathogens in response to environmental drivers. The manuscript is well written. However, there are points need to be revised. In general, the analysis method is too simple. I suggested the author add new analysis methods such as random forest, structural equation models, and others to unveil the influence of different environmental parameters.
In the introduction, there are too many paragraphs. The author should highlight the critical scope of the present work and introduce the background.
In Tables 2 and 3, the author does not do a significant difference analysis of the correlation between VG copy numbers and water parameters.
In Figures 2 and 3, it is tough to understand why the author didn’t add the error bars. The mentioned figures focused on gene copy number and time. Please explain the reason.
The information in Figures 4, 5, and 6 is hard to understand. What is Group 1? What do the points mean? Please provide more details.
In Figure 7, I think the result is too weak for further analysis.
Comments on the Quality of English Language
no comments
Reviewer 3 Report
Comments and Suggestions for Authors
In the manuscript entitled, Temperature, pH, and Salinity as Environmental Drivers for Divergence Among Harveyi Clade Pathogens and First Evidence for Vibrio harveyi and Vibrio campbellii in Southeastern USA Waters, the authors, Andrei L. Barkovskii and Cameron Brown, emphasized data regarding the presence of the Vibrio Harveyi clade fish and shellfish pathogens through the presence of six virulence genes within the water and sediments from three Georgia (USA) cultured clams and wild oyster grounds.
The article is well written but additional experiments need to be done.
It is important to prove the presence of Vibrio spp. by microbiological techniques when selective media are used. Then, based on the DNA extracted from the isolated presumptive colonies, the PCR and qPCR assays targeting both the virulence, and the specie-specific genes should be addressed. In this way, the results will be much clear regarding the spatial distribution of the water/ sediment samples containing Vibrio spp. isolates and the correlation between the number of isolates and the copy number of the virulence genes.
The results presented in table 2 and 3 are difficult to be followed. It is important to present in an individual table just the water/sediment parameters for all the samples collected (which was the water temperature, or turbidity etc. at the moment of collection?)
It would be better to present the correlation between the number of positive samples for Vibrio spp. /number of the virulence gene detected and the water/ sediment probes.
So, the idea is to corelate the VG copy number with the number of the colonies identified as Vibrio spp. and that were isolated from the site 1,2 or 3 in the respective month.
Please, specify the device used to collect the water/ sediment from the three sites.
Please, show the DNA electrophoresis image with the PCR products (toxR, luxR, srp, vhha, vhp, vhh, and rpoA).
Reviewer 4 Report
Comments and Suggestions for Authors
See the attached file.

Round 2
Reviewer 2 Report
Comments and Suggestions for Authors
no any comments
Reviewer 3 Report
Comments and Suggestions for Authors
Dear authors,
I agree with the publication in the present form.
Best regards,